# TipSegNet: Fingertip Segmentation in Contactless Fingerprint Imaging

**DOI:** 10.3390/s25061824

**Published:** 2025-03-14

**Authors:** Laurenz Ruzicka, Bernhard Kohn, Clemens Heitzinger

**Affiliations:** 1Faculty of Physics, TU Wien, 1040 Vienna, Austria; 2Digital Safety & Security, Data Science & Artificial Intelligence, Austrian Institute of Technology, Giefinggasse 4, 1210 Vienna, Austria; bernhard.kohn@ait.ac.at; 3Center for Artificial Intelligence and Machine Learning (CAIML) and Department of Computer Science, TU Wien, 1040 Vienna, Austria; clemens.heitzinger@tuwien.ac.at

**Keywords:** fingerprint, contactless, biometrics, segmentation

## Abstract

Contactless fingerprint recognition systems offer a hygienic, user-friendly, and efficient alternative to traditional contact-based methods. However, their accuracy heavily relies on precise fingertip detection and segmentation, particularly under challenging background conditions. This paper introduces TipSegNet, a novel deep learning model that achieves state-of-the-art performance in segmenting fingertips directly from grayscale hand images. TipSegNet leverages a ResNeXt-101 backbone for robust feature extraction, combined with a Feature Pyramid Network (FPN) for multi-scale representation, enabling accurate segmentation across varying finger poses and image qualities. Furthermore, we employ an extensive data augmentation strategy to enhance the model’s generalizability and robustness. This model was trained and evaluated using a combined dataset of 2257 labeled hand images. TipSegNet outperforms existing methods, achieving a mean intersection over union (mIoU) of 0.987 and an accuracy of 0.999, representing a significant advancement in contactless fingerprint segmentation. This enhanced accuracy has the potential to substantially improve the reliability and effectiveness of contactless biometric systems in real-world applications.

## 1. Introduction

Biometric identification systems have become increasingly important in various security and authentication domains, due to their reliability and uniqueness. Among these systems, the fingerprint modality stands out as one of the most widely adopted and trusted methods. Fingerprints offer a unique pattern of ridges and valleys that are consistent over an individual’s lifetime [1], making them an ideal biometric trait for personal identification and verification.

In recent years, the adoption of contactless fingerprint sensors has gained momentum, driven by the need for more hygienic, user-friendly, and versatile biometric solutions. Contactless fingerprint systems eliminate the need for physical contact with the sensor, thereby reducing the risk of transmitting infectious diseases, which is a critical advantage in the post-pandemic world. Moreover, these systems are more adaptable to various use cases, including mobile devices, public terminals, and high-security environments [2,3,4,5], where user convenience and safety are paramount.

For contactless fingerprint sensors to be effective, accurate detection and segmentation of the fingertip from the recorded images are essential. This segmentation process is crucial for several downstream tasks, such as pose correction [5,6] and feature-based matching [7], which require an accurate mask of the fingertip region to separate the area from the background. The segmentation performance directly impacts the overall accuracy and reliability of the system. Traditionally, fingertip segmentation methods have focused mostly on segmenting a single fingertip against the background, and they have relied on various techniques, including color- or brightness-based [3,8], machine learning-based [9,10], and shape-based approaches [11]. Color-based methods utilize the distinctive skin tone of fingertips, machine learning-based methods leverage machine learning algorithms to identify fingertip regions, and shape-based methods focus on the geometric properties of the fingertip.

However, in this scenario, a fingertip detection algorithm is required to first detect and classify the fingers in the image of the user’s hand. More modern approaches skip this step by directly segmenting fingertips from the hand image [10,12].

Our study follows this path and introduces a novel deep learning framework called TipSegNet, based on the ResNeXt family [13] and the Feature Pyramid Network (FPN) [14] model architecture, which surpasses the current state-of-the-art (SOTA) techniques for fingertip segmentation in contactless fingerprint images.

### 1.1. Related Work

Significant progress in fingerprint recognition frameworks has allowed us to shift from traditional contact-based methods to more advanced contactless techniques. This introduced new challenges to the process, such as fingertip segmentation from varying poses and challenging lighting and background conditions.

The process of using deep learning for object segmentation is well established, as can be seen by the work of Garcia et al. [15] and Ghosh et al. [16]. Moreover, in the field of biometrics, deep learning has significantly advanced contactless fingerprint segmentation. Murshed et al. [12] used convolutional neural networks (CNNs) to improve segmentation accuracy and robustness by training on large datasets to identify and detect fingerprint regions under varying conditions. However, instead of calculating a pixel-wise segmentation mask, they predicted a rotated bounding box around the fingertip region. This can be satisfactory for some applications; however, if detailed contour information is required, another segmentation algorithm has to be used on top of the predictions.

The work of Ruzicka et al. [5] improved compatibility between contact based and contactless fingerprint capture modalities using pose correction [6] and unwarping techniques [17], requiring and introducing a novel deep learning approach for fingertip segmentation in the process. They introduced a network architecture based on the U-Net design [18] and compared it to the network architectures of EfficientNet [19] and SqueezeNet [20]. However, in their work, they used an object detection model first to detect the fingertip bounding boxes, similar to the work of Murshed et al. in [12], and then they used the segmentation models on single finger images to separate the finger from the background and to determine the exact fingertip region of the fingerprint on the finger.

Kauba et al. [3] explored smartphone-based fingerprint acquisition, emphasizing various segmentation techniques to isolate fingerprints from the background, facilitating an effective comparison against contact-based datasets with low-latency color-based methods. They explored skin-color-based segmentation, Gaussian mixture model background subtraction and multiple deep learning approaches: Mask R-CNN [21], Deeplab [22], Segnet [23] and HRNet [24].

Priesnitz et al. [25] discussed the implementation of contactless fingerprint systems on mobile platforms using fast Otsu thresholding for fingertip segmentation. This approach can only separate the hand or the finger from the background, and it cannot differentiate different fingers.

Priesnitz et al. [10] use DeepLabv3+ to predict feature points on the hand and then fully segment the fingertips from hand images using circular areas constructed from the feature points of the hand. They return a detailed, pixel-wise segmentation mask of the input hand image.

Parallel to the development of new algorithms in the field of biometrics, the field of computer vision developed new concepts for deep learning model design. Feature Pyramid Networks (FPNs), for example, were introduced by Lin et al. [14] in 2017 to address the challenge of detecting objects at different scales in images, and they are used in combination with a feature extraction backbone. Traditional CNNs struggled with scale variance, often requiring multiple models or image resizing techniques to detect small and large objects effectively. FPNs solve this by creating a pyramid of feature maps that leverage both bottom-up and top-down pathways, with lateral connections enhancing the feature hierarchy at each level. In the biometric field, FPNs have been utilized for various tasks, such as in facial recognition systems [26], cloth segmentation for soft biometrics [27] or iris recognition [28].

For feature extraction, ResNet and ResNeXt are noteworthy. ResNet [29], known for introducing residual connections, addresses the vanishing gradient problem present in early stages of deep network design. It enables the construction of much deeper models that form the backbone of many state-of-the-art systems in image analysis. Building on ResNet, ResNeXt [13] introduces the concept of cardinality through aggregated residual transformations. This enhancement increases the model’s flexibility and scalability, allowing it to capture complex features more efficiently. In the biometric community, ResNeXt was, for example, used for ear image classification [30] or fingerprint identification [31].

Although ResNeXt and FPN architectures have seen individual use in various biometric applications, our extensive literature search indicates that their combination has not been previously explored for multi-fingertip segmentation from whole-hand images.

Several studies have enhanced contactless fingerprint recognition performance through improved fingertip segmentation. Labati et al. [32] used neural networks to address perspective distortion and rotational variations for more accurate fingerprint matching, relying on accurate segmentation. Tan et al. [6] refined minutiae extraction and matching by addressing pose variations, similarly requiring precise segmentation for the calculation of finger geometry. Chowdhury et al. [33] reviewed deep learning methodologies, emphasizing robust segmentation’s importance in recognition.

### 1.2. Contribution

Our main contribution can be summarized via three points:Novel Model Design creating TipSegNet: Utilizing transfer learning to create a ResNeXt-101-based feature extractor with a FPN-like decoder design for segmenting fingertips in hand images.Extended Data Augmentation: Augmenting the dataset with various transformations, such as perspective change, resizing and cropping, and solarization, thereby improving the model’s robustness to variations in contactless fingerprint recordings, specifically addressing changes like varying lighting conditions, different finger poses, and reducing overfitting in the training process.Comparison with SOTA: Comparing our model against established, traditional, and state-of-the-art methods, leading to the demonstration of superior segmentation performance in both cases.

## 2. Methods

In this section, we detail the methodology used for fingertip segmentation in contactless fingerprint images. In contrast to single-finger segmentation techniques, as in [5], this framework directly extracts the fingertip region of interest from the input hand image, removing the object detection step from the process.

### 2.1. Segmentation Using Deep Learning

#### 2.1.1. Pre-Processing

Before feeding the images into the model, it is crucial to standardize the input image size to ensure consistent weight matrix dimensions and stable learning.

Pre-processing of the extracted fingertips for test data involves only rescaling the images to 224×224 pixels. For training data, pre-processing includes rescaling to 224×224 pixels as well, but it also involves the application of various augmentation techniques. These augmentations are applied only with a probability of 50% and in a random order. The augmentation techniques are
Resize + Crop: The image is randomly cropped to a region of a size of 0.75 to 1 times the original size, with an aspect ratio of between 0.9 and 1.1 of the original image, before it is padded to 224×224 pixels.Rotation: The image is randomly rotated with an angle ranging from −60 to 60 degrees.Perspective Change: This technique simulates random changes in the viewpoint by distorting the image accordingly.Gaussian Blur: A Gaussian blur is applied to the image, simulating various degrees of focus and sensor noise.Solarize: This technique inverts all pixel values above a certain threshold. It creates high-contrast images and simulates the ridge-valley inversion [34].Posterize: The number of bits used to represent the pixel values is reduced, decreasing the number of possible shades of gray in the image. This simplification simulates low-contrast recording, where the background is hard to separate from the fingertips.Histogram Equalization: This method adjusts the contrast of the image by spreading out the most frequent intensity values. It is a common enhancing technique used to improve the visual appearance and downstream performance of fingerprints.

To improve the model’s ability to generalize from the training data, we employed several augmentation techniques. To assess the impact of these augmentations on performance, we conducted an ablation study. This involved training the model three times: once with the full augmentation pipeline, once without any augmentation, and once with minimal augmentation (reducing the strength of all augmentation operations).

Figure 1 displays four training set examples with applied augmentations. The images demonstrate the effects of the posterize augmentation, with the first image also featuring a resize and crop function. The second and third images illustrate perspective changes, while the third image further incorporates a rotational augmentation.

#### 2.1.2. Architecture

Our TipSegNet combines the structure of an FPN and utilizes the ResNeXt 101 32×48d architecture as a backbone. We make use of transfer learning in the backbone by starting the training with a pretrained ResNeXt 101 32×48d model instance. Pretraining was carried out on the Instagram dataset introduced by [35]. Our framework is build in PyTorch version 2.4, using the Segmentation Models framework by [36] as a starting ground.

##### ResNeXt Family

ResNeXt enhances the traditional ResNet by introducing a concept known as cardinality, which determines the number of parallel paths within each residual block [13]. We chose the ResNeXt 101 32×48d variant, which has a cardinality of 32, i.e., 32 parallel paths of convolutional layers that are concatenated together at the end of the block. This allows the network to capture a wider range of feature representations. We choose ResNeXt over ResNet because its ability to handle complex feature interactions makes it particularly well suited for challenging image recognition and segmentation tasks.

The ResNeXt architecture used in this work can be grouped into an initial part, four main layers and the finalizing part. The four main layers are depicted in blue in Figure 2. The initial part reduces the input dimension via convolutions with a stride of two and max pooling, also with a stride of two. Following the initial part is the main part with its four cardinal, residual blocks. In the first layer, a cardinal, residual block is repeated three times. In the second layer, the block is repeated four times. In the third layer, the block is repeated 23 times. Finally, the fourth main layer consists of three repeated cardinal, residual blocks. Following the main layers is the global average pooling as well as the fully connected output layer and Softmax, which is removed in this work, because the output of each of the main layers is used by the decoder part, the FPN, to create the segmentation mask output.

##### FPN and Feature Hierarchy

The FPN architecture enhances feature extraction by utilizing a top–down pathway and lateral connections. The model constructs a multi-scale feature hierarchy by progressively upsampling and merging high-level semantic features with lower-level features from earlier layers. This can be seen in Figure 2, where the four main layers of the ResNeXt architecture are symbolized using the blue color, and they depict the top–down pathway of the model. At each layer, the FPN creates a prediction, depicted with the yellow layers. Those predictions are made independently of each other. Those are then upscaled such that each of the output prediction layers matches the segmentation head dimensions, resulting in the checkerboard planes in Figure 2. Finally, the predictions are added together and fed into the segmentation head, which produces the model’s output.

##### Model Parameters and FLOPs

The model’s components, including the encoder, decoder, and segmentation head, each contribute to the overall parameter count and computational complexity. As seen in Table 1, the majority of the computation is conducted by the encoder, which also holds the most trainable parameters. To put the numbers into perspectives, 826 million trainable parameters of the encoder backbone are less than the 1843 million trainable parameters of the vision transformer ViT-G (or ViT-22B with 21743 million parameters) [37], but significantly more than the 24 million trainable parameters of a ResNet-50 or 59 million trainable parameters of a ResNet-152.

In order to investigate the effect our novel model architecture has on the performance, we conducted an ablation study comparing our backbone choice with three smaller backbones. We exchanged the ResNext-101 backbone with a ResNet-34, ResNet-50 or ResNet-101 model and trained the model for around 850 epochs each. All three backbones are smaller in terms of parameters and therefore also require less computations to be trained and run. Table 2 shows the number of parameters and FLOPs for the different choices.

#### 2.1.3. Training and Hyperparameter

Our model is trained using the Jaccard loss function [38], also known as the intersection over union (IoU) loss, which is particularly effective for segmentation tasks. Jaccard loss is defined as(1)$$\mathrm{Jaccard}\;\mathrm{Loss}=1-\frac{\left|A\cap B\right|}{\left|A\cup B\right|}$$
where *A* is the predicted segmentation mask and *B* is the ground truth mask. The ∩ operator describes the intersection and ∪ the union of the two regions, and |…| denotes taking the area of the resulting regions.

We utilize stochastic gradient descent (SGD) with minibatches and a batch size of 8 as the optimizer for training the model. The hyperparameters for SGD include a momentum of 0.9 to accelerate convergence and a learning rate of $8\times 10^{-5}$.

The model was trained for 853 epochs. During the training, we observed two major and a few minor spikes in both training and validation loss, as can be seen in Figure 3. However, the trend of both the training and the validation loss was downward. After around 10% of the run, the validation loss improved from 0.448 of the first epoch to 0.057. The final value after 853 epochs was 0.038. Although we did not observe signs of overfitting, we stopped the training run, because the outlook of further gains by continuing the training was diminishing.

The training run was conducted on an NVIDIA GeForce 3090 graphics card and took 8 days to complete.

### 2.2. Experiment

We utilize 220 manually annotated hand images from the dataset used in [5], with the addition of an in-house dataset consisting of 2016 labeled hand images recorded using a smartphone using similar recording settings. The additional in-house dataset was captured to increase the number of training examples. The data consist of hand recordings against various background scenarios. The data are split into 1788 images for training, 224 images for testing and 224 images for validation. All metrics are calculated on the test set, which was not shown to the model during training. Although the current dataset of 2236 images is sufficient for demonstrating the effectiveness of TipSegNet, we acknowledge that larger datasets are beneficial for further improving generalization and robustness. Future work will involve expanding the dataset significantly, incorporating more diverse hand images and variations in capture conditions.

The dataset annotations for the 2016 hand images from the in-house dataset were performed by a team of four people, while the annotations for the 220 annotated hand images from the dataset used in [5] were completed by a single person.

In our multi-class segmentation task, an averaging strategy is necessary for calculating the metrics over all classes. We employ micro-averaging for all metrics. Micro-averaging involves aggregating all true positives, false positives, true negatives, and false negatives across all classes before computing the overall metric. This approach ensures that each instance, regardless of its class, contributes equally to the final metric, making it particularly suitable when class distribution is imbalanced.

Moreover, we report the mean intersection over union (mIoU) metric, which is analogous to the Jaccard index, defined as the intersection of the predicted segmentation and the ground truth, divided by their union. Additionally, we report the accuracy.

## 3. Results

This section presents the results of our segmentation model, comparing its performance against state-of-the-art (SOTA) segmentation models.

The algorithms listed in Table 3 can be categorized into three groups. The first group includes approaches capable of segmenting a fingertip from an image of the entire hand but only with the following two classes: fingertip and background. In other words, they do not differentiate between different fingers. Therefore, a secondary detection stage is required. The second group represents approaches that also segmented the fingertip from an image of the entire hand; however, they use a different detection class for each of the fingers, removing the need for a secondary detection step. The third group consists of methods that segment a single finger from its background, thus requiring a detection step to prepare the data for segmentation.

In the more challenging task of segmenting the fingertip from the whole hand, eight SOTA results, apart from this work, are reported in Table 3. The first, proposed by Priesnitz et al. [10], employs Otsu adaptive thresholding to separate the hand from the background. However, since this method is limited to distinguishing an area from its background, it does not perform further fingertip detection or segmentation and therefore falls into the first group. Similar approaches to this were utilized by Kauba et al. in [3], where color histograms and Gaussian mixture models were used to segment the background of the image from the hand image. Another result presented by Priesnitz et al. in [10] makes use of the DeepLabv3+ framework. This approach predicts feature points of the hand, such as the edge of the fingertip and knuckle positions, which are then used to segment the fingertip region in the hand image.

Additionally, Kauba et al. [3] also implemented and tested four different deep learning segmentation models that are capable of segmenting the hand from its background and also segmenting the fingertips from the hand image, moving them into the second group. Those are Mask R-CNN [21], Segnet [23], HRNet [24] and DeepLab [22]. These provided the best comparison to our new approach.

In the third group, segmentation focuses solely on isolating the finger from the background, without the need for finger detection or type assignment. The first approach by Lee et al. in [39] utilizes both color and texture information in a region-growing method to segment the finger from the background. However, this method segments the entire finger, not specifically the fingertip region. The next approach, proposed by Raghavendra [40], divides the problem of fingertip segmentation into two steps. First, the entire finger is segmented from the background, similar to [39]. Then, the segmentation mask is reduced to the fingertip region using a process called scaling. Combining finger segmentation with fingertip segmentation yields an accuracy between 0.922 and 0.955, depending on the smartphone camera used. Finally, three deep learning models, U-Net, EfficientNet, and SqueezeNet, presented by [5], focus on segmenting the fingertip region from a single finger using deep learning techniques.

By integrating a Feature Pyramid Network (FPN) with a ResNeXt-101 32×48d backbone and leveraging an extensive augmentation framework, our approach achieved an accuracy of 0.999 and a mean intersection over union (mIoU) of 0.987. Additionally, we measured the F1 score to be 0.993 and the F2 score to be 0.993 as well. These results not only exceed the state-of-the-art (SOTA) performance for the challenging task of fingertip segmentation from a hand image but also outperform the SOTA methods for single-finger segmentation.

In Figure 4, we present four exemplary segmentation results. The model identifies nine classes, with Class 0 representing the background (the area surrounding the fingertips). Class 1 corresponds to the left index finger fingertip, Class 2 to the left middle finger fingertip, and so on, with Class 5 denoting the right index finger fingertip up to Class 8, which denotes the right little finger’s fingertip.

### 3.1. Failure Case Analysis

While our model achieves high overall performance, a thorough analysis of failure cases is crucial for understanding its limitations and identifying areas for improvement. We conducted a failure case analysis on both the validation and test sets. Surprisingly, only one significant failure case was found in each set, highlighting the model’s robustness.

Figure 5 shows the two identified failure cases. Figure 5 depicts a case from the validation set where the hand is partially out of bounds, with only portions of the fingertips being visible. This presents a significant challenge to the model, as it is trained to segment fingertips within the context of a complete hand. As a result, the model struggles with the segmentation of the ring and little fingers, misclassifying a considerable portion of their areas. This suggests a limitation of the model in handling incomplete hand images, particularly when critical features (the full finger context) are missing.

Figure 5 presents a case from the test set involving a finger deformity. Here, the model performs remarkably well overall, but a small number of pixels within the fingertip region are misclassified. As noted, this type of minor error is easily correctable using morphological operations (e.g., closing or filling) in a post-processing step. This case indicates that while the model is generally robust to finger shape variations, extreme deformities can still pose a minor challenge.

These two examples represent the most significant failure modes observed in our evaluation. The out-of-bounds hand highlights the importance of proper image acquisition and framing. The finger deformity case, while less problematic due to the ease of post-processing correction, suggests that incorporating more examples of such variations in the training data could further improve robustness. Overall, the scarcity of failure cases underscores the model’s strong performance, but these examples provide valuable insights for future refinement and development.

### 3.2. Data Augmentation Ablation

The results for the data augmentation ablation study, as shown in Table 4, indicate that while the augmentation did not substantially alter the top-line performance metrics (accuracy, F1, IoU), it played a crucial role in improving the model’s robustness and generalization capabilities, as evidenced by the difference in the training loss. The training loss of the augmentation free run reached the lowest minima (0.0092) and had only minor fluctuations around that minima. This implies that the model was able to extract nearly all the information for the learning signal. Therefore, it does not have further potential for improvement. The loss of the minimal augmented model (0.0262) and, even more, the loss of the fully augmented model (0.056) indicate that the images presented to the model, with the addition of the augmentations, prove to be harder to segment. This increased difficulty implies that further training on augmented data could potentially enhance the model’s generalization ability.

### 3.3. Model Ablation

The results of the model ablation study, detailed in Table 5, reveal the performance of different backbones within our segmentation framework. As the table illustrates, all tested models achieved remarkably high performance metrics, with accuracy scores nearing 0.999 and IoU scores around 0.98. These metrics are nearly saturated, being so close to the optimal value that further improvements become increasingly challenging to obtain. It suggests that we are approaching the upper limits of what is achievable with current methodologies and datasets for this particular task.

## 4. Discussion

Our study presents a significant advancement in the segmentation of fingertips in contactless fingerprint imaging by leveraging a novel deep learning approach with extensive data augmentation. In this discussion, we analyze the implications of our findings, the performance of our proposed model, and potential limitations and future directions.

The results demonstrate that our TipSegNet substantially outperforms all other advanced segmentation models. Specifically, our model achieved an accuracy of 0.999, a mIoU of 0.987 and an F1 score of 0.994. These metrics underscore the model’s robustness and precision in segmenting the fingertip regions from contactless fingerprint images. These results are particularly notable, because TipSegNet successfully segments all four fingertips directly from a hand image, a more challenging task than single-finger segmentation. This eliminates the need for a separate finger detection step, streamlining the overall biometric process. The improvements over the SOTA models can be attributed to the combination of several factors, each providing small improvements. These factors include the following:The use of ResNeXt-101 as a backbone: ResNeXt-101, with its concept of cardinality, captures a richer set of feature representations than traditional ResNet architectures. This is crucial for distinguishing subtle differences between fingertip regions and complex backgrounds.Feature Pyramid Network (FPN) integration: The FPN effectively combines multi-scale features, allowing the model to accurately segment fingertips regardless of their size or orientation in the image. This addresses a common challenge in contactless fingerprint imaging, where the finger pose can vary significantly.Extensive data augmentation: Our comprehensive augmentation strategy, including geometric transformations and intensity adjustments, significantly improves the model’s ability to generalize to diverse real-world scenarios. This is evident from the minimal difference in performance between the training and validation sets, indicating robustness to variations in image quality and capture conditions.

The marginal differences in performance metrics across the different backbones shown in the ablation study underscore a critical observation: while larger models like ResNeXt-101 do offer improvements, the gains are small compared to the substantial increase in computational resources and training time they demand.

This observation raises an important point about efficiency versus performance in deep learning model design. Although the pursuit of state-of-the-art results often leads to the development of increasingly complex models, our findings suggest that for tasks like fingertip segmentation, where performance is approaching saturation, a more balanced approach might be warranted. Smaller, more efficient models such as ResNet-34 or ResNet-50 could offer a more practical solution, providing a good trade-off between performance and computational efficiency. Especially in resource-constrained environments or applications requiring rapid processing, these models could deliver nearly equivalent results in this framework without the overhead associated with their larger counterparts.

The ResNeXt-101 backbone and FPN decoder, while effective, significantly increase the computational complexity of our model. This complexity can lead to longer training and inference times, particularly on resource-constrained devices. Parallel computation techniques, especially leveraging the parallel processing capabilities of modern GPUs, offer a promising avenue for mitigating this issue. The inherent structure of ResNeXt, with its multiple parallel branches (cardinality), is naturally suited for parallelization across GPU cores. Similarly, the independent feature processing at different levels of the FPN can be distributed across multiple GPU threads. Frameworks and libraries such as cuDNN [41] provide highly optimized primitives for deep learning on GPUs, enabling significant acceleration of both training and inference. Efficient utilization of GPU resources can significantly accelerate both training and inference, making the model more practical for real-time applications.

Furthermore, the saturation of performance metrics highlights the need for more challenging datasets that can better differentiate between model capabilities. As we push the boundaries of what is possible with current techniques, identifying areas where models still struggle can guide future research and innovation in the field. For example, future datasets could include more diverse backgrounds, challenging lighting conditions, and variations in skin tone and texture.

Our augmentation ablation study highlights the importance of data augmentation in achieving robust model performance. Although the impact on the overall metrics was not substantial, the lower training loss observed with no augmentation suggests that it plays a critical role in preventing overfitting and enhancing the model’s ability to generalize. This is particularly important when dealing with limited training data, as is often the case in biometric applications.

The accuracy of fingertip segmentation directly influences the performance of downstream tasks such as pose correction, feature-based matching, and overall fingerprint recognition. Improved segmentation accuracy ensures that the extracted fingertip regions are precise, which enhances the reliability of subsequent processing steps. This is particularly important for contactless fingerprint systems, where variations in perspective and environmental conditions can introduce additional challenges. By accurately segmenting all four fingertips, TipSegNet provides a solid foundation for these downstream tasks, leading to more accurate and reliable fingerprint recognition.

Despite its high efficacy, the model’s complexity and parameter count (829 million parameters) may pose challenges for deployment in resource-constrained environments. Future work could focus on model compression techniques, such as pruning and quantization, to reduce the computational load without compromising accuracy. Additional possibilities include the automatic labeling of large datasets using this model, which can then be used by smaller models to train on. Furthermore, the concept of knowledge distillation [42,43] can also be used to train a smaller model while using the bigger, more capable model as a teacher.

Finally, while our model demonstrates beyond state-of-the-art performance, it is important to acknowledge that the field of contactless fingerprint recognition is rapidly evolving. Future research should explore the integration of TipSegNet with other advanced techniques, such as 3D fingerprint reconstruction and liveness detection, to develop even more robust and secure biometric systems.

### Limitations

Despite TipSegNet’s strong performance, certain limitations warrant consideration. Occlusion of fingertips by other fingers or external objects presents a significant challenge. Although the multi-scale representation provided by the FPN offers some robustness to minor occlusions, severe obscuration, where a substantial portion of the fingertip is hidden, is likely to degrade segmentation accuracy. Image quality also plays a crucial role; noise, low resolution, or inadequate lighting can negatively impact performance, particularly in resolving fine details such as precise fingertip contours. Although pre-processing techniques, such as noise reduction, contrast enhancement, and resolution upscaling as described in [44], could potentially mitigate some of these issues, inherent limitations in spatial resolution and receptive fields of convolutional layers can still hinder the discrimination of extremely fine-grained features, which is not that relevant for the task of fingertip segmentation. Furthermore, while the dataset employed in this study is of a sufficient size and includes augmentation techniques, its limited diversity in terms of finger deformities and extreme hand poses is a limitation. Finally, the computational cost associated with the ResNeXt-101 backbone and FPN decoder is substantial, potentially limiting deployment in resource-constrained environments.

## 5. Conclusions

This research introduces TipSegNet, a novel deep learning model designed for accurate, multi-finger segmentation in contactless fingerprint images. By integrating a robust ResNeXt-101 backbone with a Feature Pyramid Network and employing a comprehensive data augmentation strategy, TipSegNet surpasses existing state-of-the-art methods, achieving a mean intersection over union (mIoU) of 0.987, an accuracy of 0.999, and an F1 score of 0.994. These results demonstrate a significant advancement in the field, particularly in the challenging context of segmenting multiple fingertips directly from hand images without a separate finger detection step. Our ablation studies further highlight the effectiveness of our design choices, demonstrating that while larger models like ResNeXt-101 offer marginal gains, careful consideration of model size and computational resources is crucial for practical deployment. The high accuracy of TipSegNet not only provides a solid foundation for downstream biometric tasks, such as pose correction and feature matching, but also opens new avenues for creating more streamlined and user-friendly contactless fingerprint recognition systems. Future work could focus on optimizing TipSegNet for real-time processing, exploring model compression techniques to enhance efficiency, and investigating its integration with advanced biometric methodologies, including 3D fingerprint reconstruction and liveness detection. This will pave the way for the development of even more robust, secure, and widely applicable biometric authentication solutions.

## Figures and Tables

**Figure 1 sensors-25-01824-f001:**
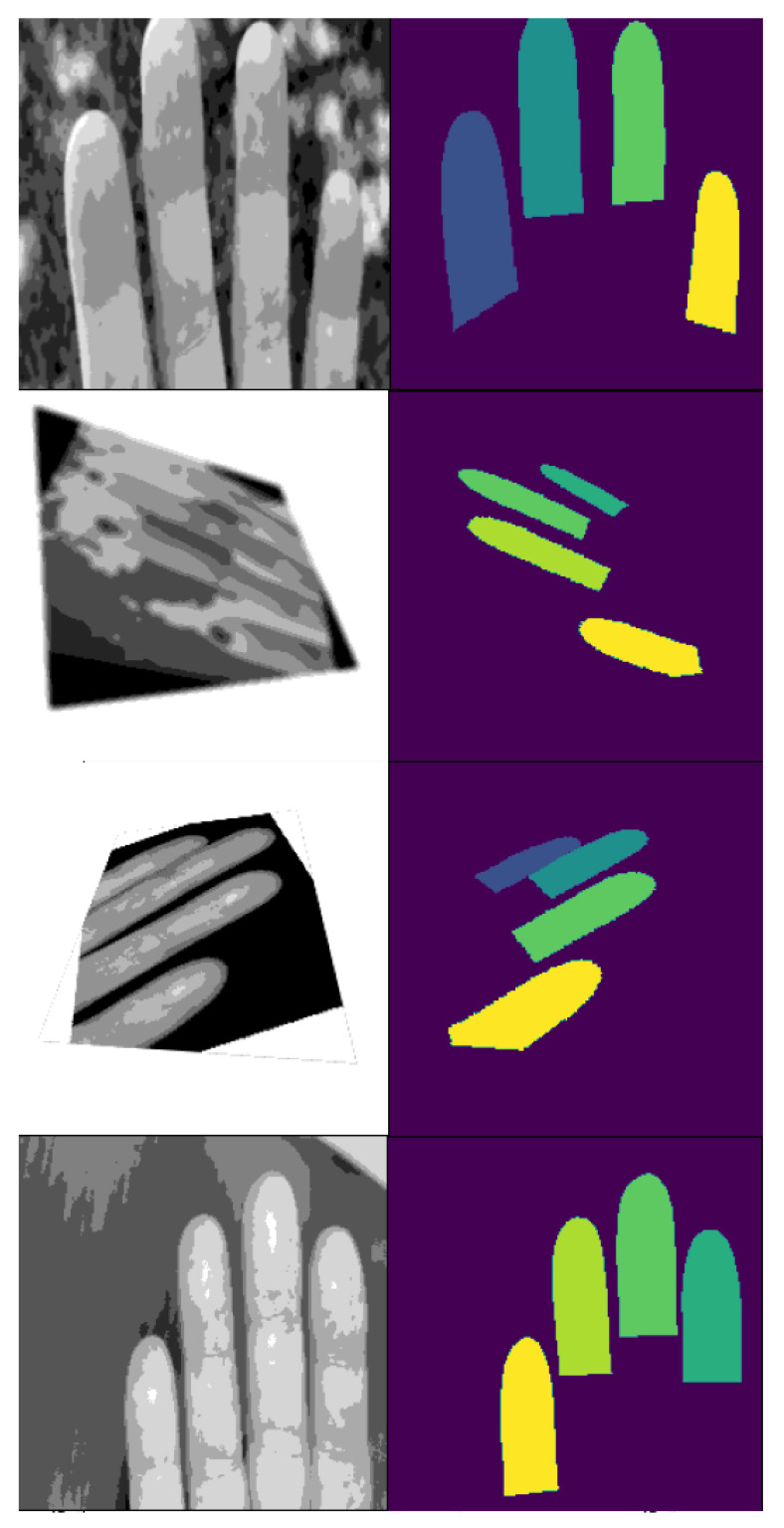
Examples from the training set with added augmentations. Input images as shown to the model during training on the left and their corresponding labels on the right.

**Figure 2 sensors-25-01824-f002:**
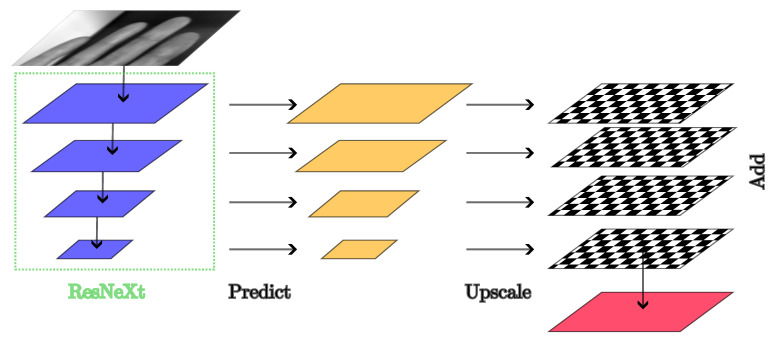
Model architecture. The ResNeXt part is encircled by the dashed, green line, and the output of its four main layers is used by the FPN to generate the multi-scale predictions (yellow), which are then upscaled (checkerboard pattern) before being summed together to create the input to the segmentation head (red).

**Figure 3 sensors-25-01824-f003:**
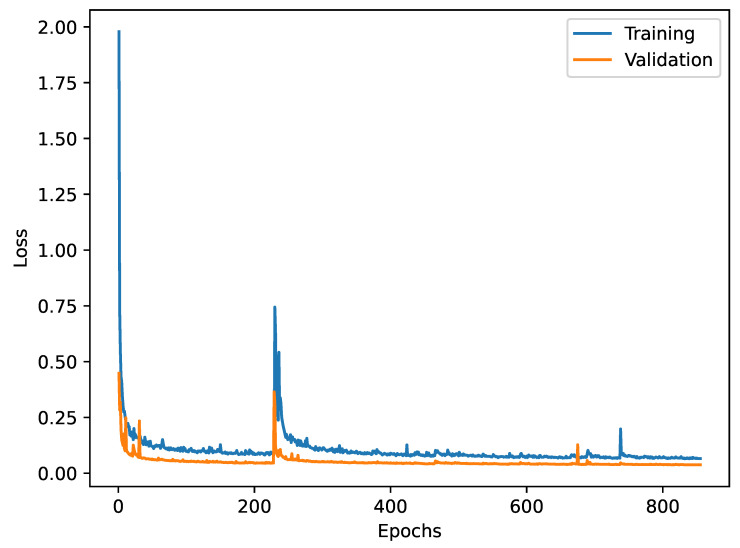
Training and validation loss over training epochs.

**Figure 4 sensors-25-01824-f004:**
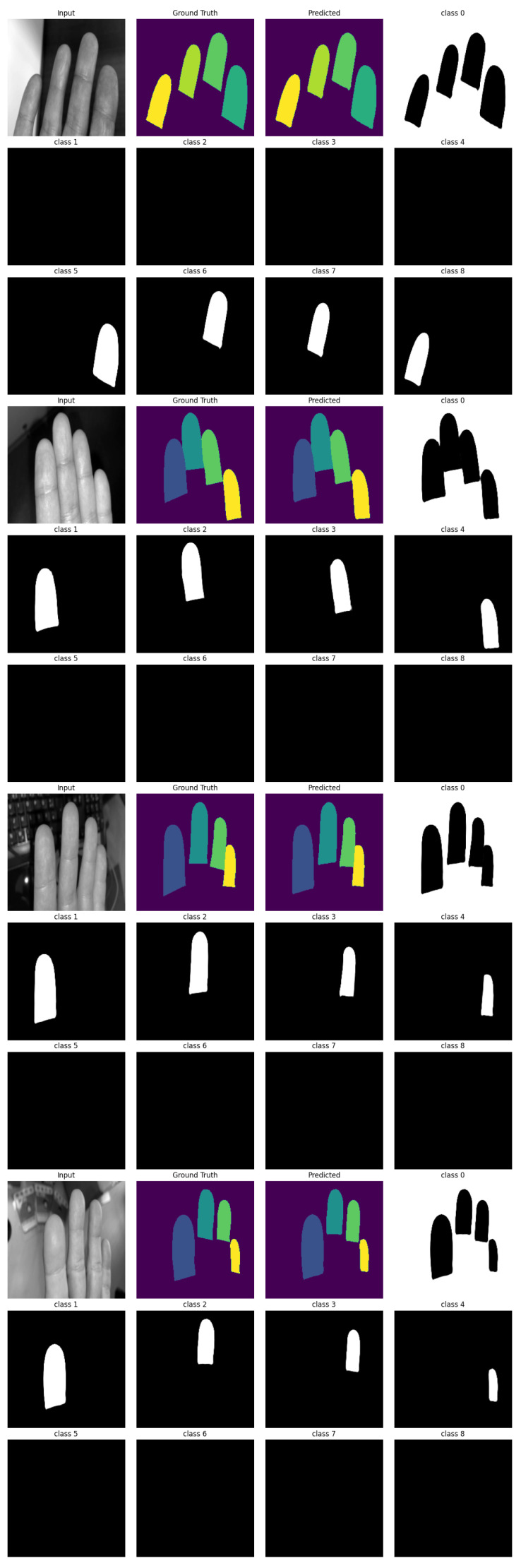
Four exemplary segmentation results from the validation set. Class 0 describes the separation of the fingers from the background, Class 1 the left index finger, Class 2 the left middle finger and so on.

**Figure 5 sensors-25-01824-f005:**
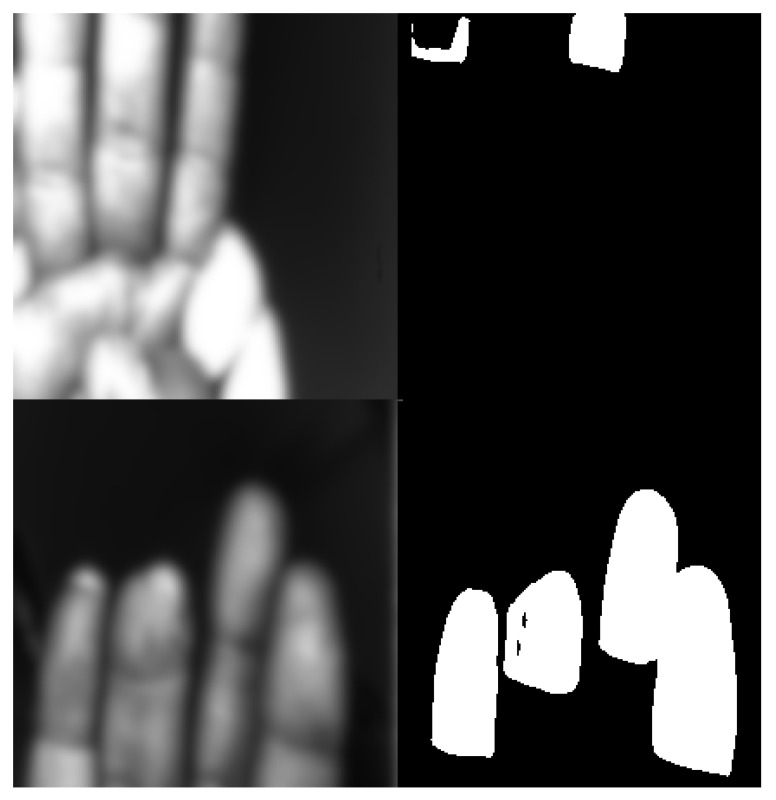
Failure case examples with blurred input images for privacy reasons. (top) Example from the validation set with an out-of-bounds hand, leading to mis-segmentation of the ring and little fingers. (bottom) Example from the test set with a finger deformity, resulting in a small number of misclassified pixels within the fingertip region.

**Table 1 sensors-25-01824-t001:** Number of parameters and floating point operations (FLOPs) for the different model parts and the whole model.

Model Part	Parameters	FLOPs
Encoder	$8.264\times 10^{8}$	$1.231\times 10^{12}$
Decoder	$2.608\times 10^{6}$	$1.508\times 10^{10}$
Segmentation Head	1161	$4.335\times 10^{7}$
Total	$828.965\times 10^{8}$	$1.246\times 10^{12}$

**Table 2 sensors-25-01824-t002:** Number of parameters and floating point operations (FLOPs) for ResNet-34, ResNet-50 and ResNet-101.

Model	Parameters	FLOPs
ResNet-34	$2.315\times 10^{7}$	$4.163\times 10^{10}$
Encoder	$2.128\times 10^{7}$	$2.883\times 10^{10}$
Decoder	$1.871\times 10^{6}$	$1.276\times 10^{10}$
ResNet-50	$2.611\times 10^{7}$	$4.763\times 10^{10}$
Encoder	$2.350\times 10^{7}$	$3.251\times 10^{10}$
Decoder	$2.608\times 10^{6}$	$1.508\times 10^{10}$
ResNet-101	$4.510\times 10^{7}$	$7.753\times 10^{10}$
Encoder	$4.249\times 10^{7}$	$6.242\times 10^{10}$
Decoder	$2.608\times 10^{6}$	$1.508\times 10^{10}$

**Table 3 sensors-25-01824-t003:** Segmentation scores for various SOTA methods as reported in the corresponding publications, taken with the reported decimal places. The topmost entries until the horizontal line are methods that work with hand images as inputs, while the others only work on a single finger. **Bold** values indicate the highest scores, for both the whole hand and the finger-only group. *Italic* indicates our approach.

	Approach	mIoU	Accuracy
Group 1	Otsu [10] ^1^	0.92	-
	Color Histogram [3] ^1^	0.38	-
	Gaussian Mixture [3] ^1^	0.31	-
	Mask R-CNN [3]	**0.96**	-
	DeepLabv3+ [10]	0.95	-
Group 2	Segnet [3]	0.90	-
	HRNet [3]	0.85	-
	DeepLab [3]	0.93	-
	*TipSegNet*	**0.99**	**1.00**
Group 3	Color & Texture [39] ^2^	-	**0.99**
	Mean Shift [40] ^2^	-	0.92–0.96
	U-Net [5] ^2^	**0.91**	0.98
	EfficientNet [5] ^2^	0.50	0.88
	SqueezeNet [5] ^2^	0.86	0.96

^1^ Segmentation conducted only for the hand area; no fingertip detection. ^2^ Segmentation conducted only for a single finger.

**Table 4 sensors-25-01824-t004:** Augmentation ablation results, where no augmentation describes the results for the model trained without augmentations; the results for the model with only minor augmentation and data augmentation; and the results for the model trained with full data augmentation.

	Accuracy	F1	IoU	Recall
no augmentation	0.999	0.994	0.987	0.994
minimal augmentation	0.998	0.993	0.985	0.993
data augmentation	0.999	0.994	0.987	0.994

**Table 5 sensors-25-01824-t005:** Ablation results for different model backbones.

Model	Accuracy	F1	IoU	Recall
ResNet-34	0.998	0.990	0.981	0.990
ResNet-50	0.997	0.988	0.976	0.988
ResNet-101	0.998	0.992	0.984	0.992
ResNeXt-101	0.999	0.994	0.987	0.994

## Data Availability

Data are contained within the article.

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
