# Peer review of "TipSegNet: Fingertip Segmentation in Contactless Fingerprint Imaging"

_sensors, 2025, doi:10.3390/s25061824_

Round 1

Reviewer 1 Report

Comments and Suggestions for Authors

Overall Impression

The paper presents a well-structured and technically sound study on contactless fingerprint segmentation using deep learning. The authors introduce TipSegNet, ResNeXt-101 and FPN, and demonstrate its superiority over existing methods. The methodology is well-explained, results are compelling, and the comparisons with state-of-the-art approaches are thorough. The writing is clear and informative, making this work a valuable contribution to the field of contactless biometrics.

Abstract:

1. Effectively summerized the research problems, methodology, and results.

2. Minor correction: Consider adding a brief statement about the dataset size used for training and testing to give readers a sense of the model's generalizability.

Introduction

1. The introduction clearly defines the problem and its relevance in biometric security and have good motivation

2. Good relevant paper discussed.

Contribution

1. The contributions are clearly listed and effectively highlight the novelty of TipSegNet.

Minor Correction:

The contribution related to "Extended Data Augmentation" is valuable, but it would be helpful to briefly mention how augmentation improved robustness (e.g., handling different lighting conditions).

Methods:

  • 1. The methodological framework is well-structured and detailed, making replication feasible.
  • 2. The choice of ResNeXt-101 and FPN is justified based on their ability to handle scale variations.
  • 3. The pre-processing steps are well described and enhance model generalizability.
  • Minor Correction
  • The figure describing the model architecture could benefit from a slightly more detailed annotation of the different components.   
  • Experiment
  • The dataset composition and experimental setup are well described.
  • The use of micro-averaging for handling multi-class segmentation is a good choice.
  • The decision to split the dataset into training, validation, and test sets is appropriate.
  • Results
  • The results clearly demonstrate the model’s superior performance over existing methods.
  • The use of multiple evaluation metrics (accuracy, mIoU, F1-score) provides a robust assessment.
  • The visualization of segmentation results helps validate the model's effectiveness.
  • Model Ablation
  • The comparative analysis of different backbones is highly informative.
  • The trade-off discussion between model complexity and segmentation performance is well addressed.

Reviewer 2 Report

Comments and Suggestions for Authors

 The research design of the TipSegNet study demonstrates appropriateness for its objectives, effectively integrating a ResNeXt-101 backbone with a Feature Pyramid Network (FPN) for fingertip segmentation. The study benefits from direct segmentation, eliminating the necessity for separate finger detection, and employs a comprehensive data augmentation strategy to enhance robustness. 

Weakness

The authors claim three main contributions in their paper, but their work appears to lack novelty. They propose designing a novel model using ResNeXt-101 + FPN. However, ResNeXt-101 as a backbone and FPN (Feature Pyramid Network) decoders are already well-established and widely used architectures. The combination itself is not novel, as similar architectures have been used for various segmentation tasks. Additionally, using transfer learning with these components is a standard practice. The actual novelty would depend on specific architectural modifications or adaptations made for fingertip segmentation. However, it is unclear whether any specific architectural modifications were made to either ResNeXt-101 or FPN to improve results on their dataset.

The dataset used in the study is very small, and the authors need to expand its size. There is also insufficient discussion regarding the quality and consistency of the annotations. The paper does not specify how many people participated in the annotation process or whether any tools were used to mitigate human bias. Furthermore, there is no mention of inter-subjective validation that could ensure annotation consistency, which is crucial for dataset robustness.

The augmentation techniques mentioned in the paper are all standard practices in the field. Moreover, the authors themselves note that "the augmentation had no significant impact on model performance regarding prediction quality," which calls into question the necessity of including augmentation in their pipeline.

The study does not analyze failure cases. 

Reviewer 3 Report

Comments and Suggestions for Authors

The paper has addressed a nice problem. I do agree that the fingertip detection and segmentation method that utilizes a ResNeXt-101 backbone for robust feature extraction combined with a feature pyramid network for multi-scale representation, is a powerful approach for handling complex tasks such as fingertip detection across varying poses and image qualities. However, there are still several limitations inherent to this method that the authors need to revisit during their revision:

1. ResNeXt-101 and FPN add more complexity to the network. the authors should discuss if parallel computation could be of any help by citing the below paper:

“Lattice-Boltzmann Interactive Blood Flow Simulation Pipeline,” International Journal of Computer Assisted Radiology and Surgery, Springer, vol.15, pp. 629-639, 2020.

2. Fingertips can be occluded by other fingers or objects, especially in complex hand poses. While multi-scale representations (via FPN) can handle some occlusion, there may still be challenges in accurately detecting and segmenting fingertips when they are obscured.

3. Image quality plays a significant role in segmentation accuracy. If the input images suffer from noise, low resolution, or poor lighting conditions, the model may fail to accurately detect or segment fingertips, especially in cases where fine details are important for localization.

Fingertips are small regions in the image, and the background or hand palm areas dominate the image space. This can lead to a class imbalance issue, where the model focuses more on detecting the larger regions of the image (e.g., palm or wrist) rather than the smaller fingertip regions. Could pre-processing help? Pre-processing techniques can probably serve as a foundation for better feature extraction, reducing noise, and improving the effectiveness of feature fusion and attention modules. The below papers may be cited while discussing this:

“Advancements in Deep Learning for B-Mode Ultrasound Segmentation: A Comprehensive Review,” IEEE Transactions on Emerging Topics in Computational Intelligence, vol. 8, no. 3, pp. 2126-2149, 2024

4. The model might still face challenges in discriminating small, fine-grained features, such as the precise contour of fingertips.

5. Please include the potential limitations of the paper. 

Comments on the Quality of English Language

The language can further be improved. 

Round 2

Reviewer 3 Report

Comments and Suggestions for Authors

no more comments